# High Incidence of Adverse Outcomes in Haemodialysis Patients with Diabetes with or without Diabetic Foot Syndrome: A 5-Year Observational Study in Lleida, Spain

**DOI:** 10.3390/jcm10071368

**Published:** 2021-03-26

**Authors:** Montserrat Dòria, Àngels Betriu, Montserrat Belart, Verónica Rosado, Marta Hernández, Felipe Sarro, Jordi Real, Esmeralda Castelblanco, Linda Roxana Pacheco, Elvira Fernández, Josep Franch-Nadal, Mònica Gratacòs, Dídac Mauricio

**Affiliations:** 1Department of Endocrinology & Nutrition, University Hospital Arnau de Vilanova and Santa Maria, 25198 Lleida, Spain; montserratdoria@gmail.com (M.D.); veviroce@hotmail.com (V.R.); mhernandez.lleida.ics@gencat.cat (M.H.); linda_rox@hotmail.com (L.R.P.); 2Department of Endocrinology and Nutrition, Hospital de la Santa Creu i Sant Pau, Autonomous University of Barcelona, 08041 Barcelona, Spain; 3Sistemes Renals, 2508 Lleida, Spain; abetriu@sistemes-renals.com (À.B.); mbelart@sistemes-renals.com (M.B.); 4Lleida Institute for Biomedical Research Dr. Pifarré Foundation IRBLleida, University of Lleida, 25198 Lleida, Spain; elvirafgiraldez@gmail.com; 5Department of Nephrology, University Hospital Arnau de Vilanova and Santa Maria, 25198 Lleida, Spain; Jfsarro.lleida.ics@gencat.cat; 6DAP-Cat group, Unitat de Suport a la Recerca Barcelona, Fundació Institut Universitari per a la recerca a l’Atenció Primària de Salut Jordi Gol i Gurina (IDIAPJGol), 08006 Barcelona, Spain; jordireal@gmail.com (J.R.); esmeraldacas@gmail.com (E.C.); josep.franch@gmail.com (J.F.-N.); monica.gratacos@gmail.com (M.G.); 7Centre for Biomedical Research on Diabetes and Associated Metabolic Diseases (CIBERDEM), Instituto de Salud Carlos III (ISCIII), 08041 Barcelona, Spain; 8Primary Health Care Center Raval Sud, Gerència d’Atenció Primaria, Institut Català de la Salut, 08001 Barcelona, Spain; 9Faculty of Medicine, University of Vic (UVIC/UCC), 08500 Vic, Spain

**Keywords:** diabetes mellitus, haemodialysis, diabetic foot syndrome, foot ulcer, amputation, all-cause mortality, cardiovascular mortality, survival

## Abstract

Background: We evaluated whether, in subjects receiving haemodialysis (HD), the presence of diabetic foot syndrome (DFS) was associated with increased mortality compared with subjects with diabetes mellitus (DM) without DFS and with non-diabetic subjects. Methods: Retrospective, observational study in 220 subjects followed for six years. We calculated and compared the frequency and 5-year cumulative incidence of all-cause mortality, cardiovascular (CV) mortality, CV events, major adverse CV events (MACE), and new foot ulcer (FU) or amputation. We also examined prognostic factors of all-cause and CV mortality based on baseline characteristics. Results: DM patients had a 1.98 times higher probability of all-cause mortality than those without DM (*p* = 0.001) and 2.42 times higher likelihood of CV mortality and new FU or amputation (*p* = 0.002 and *p* = 0.008, respectively). In the DM cohort, only the risk of a new FU or amputation was 2.69 times higher among those with previous DFS (*p* = 0.021). In patients with DM, older age was the only predictor of all-cause and CV mortality (*p* = 0.001 and *p* = 0.014, respectively). Conclusions: Although all-cause and CV mortality were increased on HD subjects with DM, the presence of DFS did not modify the excess risk. Additional studies are warranted to further explore the impact of DFS in subjects with DM undergoing HD.

## 1. Introduction

Diabetes mellitus (DM) is the primary cause of chronic kidney disease (CKD) and, alone or in combination with hypertension, the cause of 80% of end-stage renal disease (ESRD) cases [1]. On the one hand, the already high risk in the 10-year cumulative all-cause mortality for DM increases by 20% when accompanied by CKD [1]. The reasons for this increased mortality include the presence of significant cardiovascular disease (CVD), problems with vascular access, increased susceptibility to infectious complications, haemodynamic instability during HD due to autonomic neuropathy, and foot ulcer (FU) [2]. On the other hand, the mortality of patients requiring chronic dialysis is higher among subjects with DM, with an estimated 5-year survival as low as 30% after initiation of haemodialysis (HD) [2]. Similarly, predictors of mortality on HD patients are peripheral artery disease (PAD), presence of foot ulcers, and DM [3]. Indeed, it has been estimated that the mean survival rate of patients on HD and with foot lesions is only 1.8 years [4].

Foot ulcers are one of the pathological aspects of diabetic foot syndrome (DFS) and a chronic complication of DM, frequently associated with subsequent foot amputation and death [5]. For instance, subjects with FU are 2- to 3- times more likely to die than patients without FU, and the risk ratio for all-cause mortality is almost 2-fold higher in subjects with DM and FU than in subjects with DM only [6,7,8,9,10,11,12,13]. Of note, the incidence of DFS is much higher in patients with DM than in the general population across all categories of renal disease (i.e., not requiring replacement therapy, treated with HD or PD, and requiring renal transplantation or simultaneous pancreas and kidney transplantation) [14]. It has been estimated that up to 95% of dialysis-treated diabetic patients are at high risk of foot problems, and HD itself is an independent risk factor for foot ulceration, non-healing, and amputation [15,16,17]. About 20% of DM patients will develop a FU one year after dialysis initiation, and they also have a higher incidence of new amputations and foot-related hospitalizations [18,19,20]. Risk factors, associated with the likelihood of developing a foot ulcer or lower limb amputation, other than DM, when in maintenance dialysis, are peripheral artery disease (PAD), peripheral neuropathy (PN), and coronary artery disease (CAD) [21,22,23].

The difference in mortality rates in DM patients requiring dialysis with or without FU has not been studied by large, and results are, in some cases, inconsistent [3,24,25]. In a previous study conducted by our group, we assessed the prevalence of DFS and other associated conditions in DM patients under renal replacement followed for five months [26]. In the present study, we further evaluated whether the presence of DFS was associated with increased mortality compared with subjects with DM but not FDS and with subjects without DM, followed for six years. Moreover, we evaluated differences in CV event risk and the development of new foot ulcerations and amputations, and predictors of all-cause and CV mortality.

## 2. Materials and Methods

### 2.1. Design

This was a retrospective, observational study comparing two cohorts recruited between November 2010 and March 2011 and followed until December 2017. The first cohort consisted of subjects previously described in a cross-sectional study, namely patients with DM and CKD receiving HD as renal replacement therapy between November 2010 and March 2011. Subjects were recruited from the two dialysis centres available in Lleida’s health area, a Northeast region of Spain [26]. The other cohort, from the same geographical and clinical setting, consisted of all prevalent subjects without DM, receiving HD replacement therapy (i.e., at least for one month) up to March 2011 (identical time period as the initial cohort).

### 2.2. Studied Variables

The baseline clinical variables for the cohort of DM patients were those described for the initially recruited cohort, excluding subjects under peritoneal dialysis (*n* = 7) [26]. Briefly, information was obtained on age, gender, HD initiation date, hypertension, dyslipidemia, smoking status, retinopathy, and neuropathy. Patients underwent a detailed foot examination, and the following variables were recorded: current and previous FU, previous lower-limb amputations, and the presence of PAD [26]. For the new cohort of non-DM patients, the same data and variables (except for neuropathy and retinopathy) were extracted from the dialysis clinics’ quarterly reports, and the results of the vascular explorations were recorded in the electronic clinical history of the Detection and Treatment Unit of Atherothrombotic Diseases (UDETMA). This unit, pertaining to the Nephrology Unit of the Hospital Arnau de Vilanova (Lleida), uses the same methodology and instruments as in our previous report [27]. The baseline vascular disease history was obtained as an additional variable for both patients with or without DM and extracted from the electronic medical records after hospital discharge and the dialysis clinic’s quarterly reports. It included any of the following: a history of ischemic heart disease, cerebrovascular disease regardless of the origin, heart failure, PAD, revascularisation procedures, major or minor amputation, and surgical ulcer debridement. During the 6-year follow-up period, and based on hospital discharge data, we also recorded admissions for any cause, the last date when the patient was known to be alive and, if applicable, the date of the end of HD treatment, which could be due to lost to follow-up, renal transplant, or death. In the latter case, the cause of death was recorded as CV, non-CV, or unknown.

### 2.3. Statistical Analysis

In the descriptive analyses of the different groups (i.e., non-DM and DM with or without DFS, defined as previous or current foot ulcer or amputation [FU/A]), categorical variables are presented as absolute and relative frequency, and continuous variables expressed as means ± standard deviation (SD). The t-test was used for continuous variables and the exact Fisher test for categorical variables to assess differences between groups.

The following events were considered as outcomes: all-cause mortality, CV mortality, the incidence of CV events (coronary heart disease [CHD], cerebrovascular disease [CeVD], PAD, mesenteric ischemia, and ischemic colitis), the incidence of major adverse CV events (MACE; including non-fatal cerebrovascular event, non-fatal ischaemic coronary event, or CV death), and new ulcer or need for amputation. Each event’s frequency was calculated, and the 5-year cumulative incidence and time-to-event analysis were performed. Hazard ratios (HR) for each outcome by groups based on baseline characteristics were estimated. A standard survival analysis using the Kaplan-Meyer method was conducted to generate the curves with the estimated time-to-event. Finally, HR for each outcome, with 95% confidence intervals (95% CI), were estimated with competing-risks regression based on the Fine and Gray’s proportional subhazard model to estimate the probability of each event correctly [28]. Additionally, adjustments were made for age, gender, hypertension, dyslipidaemia, smoking, and any previous CVD. All analyses were conducted with the free software environment for statistical computing R version 3.5.3 (2019-03-11) for Windows.

## 3. Results

The characteristics of the two cohorts are shown in Table 1. Overall, 220 patients on HD were followed for a median of 5 years (1828 days). The mean age was 67.5 years and 60% were male. Of all HD patients, 38.6% (*n* = 85) had DM, with a median disease duration of 19.7 years, and most cases were type 2 DM (87.1%). Previous or current DFS was found in 35.3% of patients from the overall diabetic HD population, and clinical history of FU (previous or current) or amputation in 6.7% of subjects without DM.

The main differences between the DM and non-DM cohorts were that patients with DM had a higher prevalence of dyslipidemia and PAD (*p* < 0.001), and history of CHD and CVD (*p* = 0.011 and *p* = 0.006, respectively). In the subgroup of DM patients, we observed no differences between those with or without DFS as regards mean age, gender, type of DM, disease duration, or diabetes-related microvascular complications (neuropathy, retinopathy, and nephropathy; Appendix A). Only clinical history of CVD was more frequent among those with DFS (*p* = 0.021).

### 3.1. Outcomes at 5-Years

After five years of follow-up, 52 patients (23.6%) on HD underwent renal transplant, which was less frequent in DM patients (11.8% vs. 31.1%; *p* < 0.001). Only one patient was lost to follow-up.

At the end of the follow-up, more than half of the patients had died (*n* = 123; 55.9%). The all-cause mortality was significantly more frequent among patients with DM than without (*p* < 0.001), as were CV deaths (*p* < 0.001), CV events (*p* = 0.001), MACE (*p* = 0.001), and the incidence of FU or need of amputation (*p* < 0.001) (Figure 1a).

Among those DM patients on HD (*n* = 85), the only clinical outcome that was significantly elevated among those with DFS was the development of a new FU or the need for amputation (*p* = 0.004; Figure 1b).

The Kaplan-Meyer curves showed that, for all-cause mortality, CV mortality, and new FU or amputation, the median time to each outcome was shorter among those with DM (Appendix A). As such, patients with DM had an almost 2-fold higher probability of all-cause mortality compared with those without DM (HR = 1.98), and a more than 2-fold higher likelihood of CV mortality or new ulcer or amputation (HR = 2.42; and HR = 2.29, respectively) (Figure 2a). In contrast, the risk of an incident CV event or MACE was comparable between those with or without DM.

When only considering the DM patient cohort, the median time to all-cause death, CV death, a CV event, or a MACE was similar between those with DFS compared with those without, but it was significantly shorter for the appearance of a new FU or need for amputation among those with previous DFS (Appendix A). Specifically, the risk of a new ulcer or amputation was 2.69 times higher among those with previous DFS than those without (*p* = 0.021) (Figure 2b).

### 3.2. Predictors of All-Cause Mortality and CV Mortality

#### 3.2.1. Overall Dialysis Population

The all-cause mortality in the overall HD population was more likely among those with older age, DM, antecedents of cardiac conditions (i.e., CHD, CVD, arrhythmia, and HF), diabetes-related microvascular complications (i.e., diabetic retinopathy and neuropathy), PAD, previous FU, and both previous minor and major amputations (Table 2). When only considering deaths attributable to CV events, females were at lower risk, but previous FU and minor or major amputations were not associated with a higher likelihood of CV mortality.

#### 3.2.2. DM Cohort

Among DM patients, older age was the only risk factor associated with increased probability of all-cause and CV mortality (Table 3). Moreover, diabetic retinopathy was a predictor of increased risk for overall mortality. Thus, neither the presence of previous or current FU, nor history of minor or major amputations were predictors of an increased global or CV mortality.

## 4. Discussion

The current observational study showed that, in patients with DM undergoing HD replacement therapy, the risk of all-cause and CV mortality and incident FU or amputation was increased compared with those without DM. However, the 5-year survival probability was similar for diabetic patients with or without DFS, thus not modifying the already elevated risk for overall and CV mortality.

More than half of the patients died during our study, which is in line with previous reports showing that the 5-year survival probability in patients in maintenance HD is around 42–45% [29,30,31]. Moreover, and also in agreement with previous studies [2], the mortality rates were significantly higher among patients with DM (64% vs. 38%), with a 2-fold reduced probability of 5-year survival compared with non-diabetic subjects.

The survival of HD patients with foot lesions is very poor and estimated at 23% at five years [4]. This figure was similar in our cohort (33.3%). However, it was unexpected that the rates of all-cause and CV mortality, MACE, or CV events were similar between patients with and without DFS. Firstly, several studies have consistently shown that the rate of death, myocardial infarction, and fatal stroke in patients with DM is higher among those with FU [12,32,33,34,35]. Of note, although the main cause of mortality is CVD, a meta-analysis reported that the proportion of deaths attributable to CV causes was similar among patients with or without FU [12]. Moreover, a large study conducted in the US reported that the severity of diabetic FU at presentation predicted subsequent mortality to a greater extent than prior CVD [33]. Secondly, few studies have addressed these outcomes on HD patients with DM and concomitant FU [3,24]. Our results are not in agreement with a study conducted by Al-Thani et al., in Qatar, including 252 HD patients, where the 5-year mortality was higher among patients with diabetes and FU vs. those without FU (83% vs. 58%) [3]. Our results are also in disagreement with another study conducted by Garimella et al., in the US in 14,103 people with diabetes on dialysis, where death was more likely among patients with incident FU (*n* = 1769; 25.5% deaths) than among those at-risk (no FU during follow-up; *n* = 11,750; 19.1% deaths) [24]. The explanation for these apparent discrepancies may be related to methodological and population differences between the studies. The Al-Thani et al., study included an HD population considerably younger than ours (almost 13 years younger for those without FU, and seven years younger for those with FU), and less likely to have DM plus FU than ours (23% vs. 35.5%). Moreover, while their FU cohort had increased age, nephropathy, retinopathy, coronary artery disease, and PAD compared with the non-FU cohort, we only observed a slightly increased CVD prevalence among those with DM and FU vs. those without FU (36.7% vs. 30.9%). Regarding the Garimella et al. study, the authors included subjects with incident FU and excluded those with a prevalent ulcer at first foot-check. In contrast, we studied patients with both previous and prevalent FU (60% and 53.3%, respectively). Therefore, the populations studied are not comparable and possibly had different comorbidities, such as diabetes-related complications (e.g., micro- and macrovascular diseases). Indeed, it was previously reported that the impact of variables associated with CV risk is greater among subjects presenting their first diabetic FU and that the variables related to the risk of dying are different between those with or without a history of previous FU [36].

The risk factors influencing all-cause and CV mortality in patients undergoing HD are multiple and include DM, age, previous CVD, and haemodialysis duration [37]. Moreover, the characteristics of patients with DM in dialysis differ from those without DM: diabetic patients are older, present more CVD, and are less likely to be transplanted [38]. These differences were also observed in our study, where T2DM, older age, antecedents of cardiac conditions (i.e., CHD, CVD, arrhythmia, and HF), diabetes-related microvascular complications (i.e., diabetic retinopathy and neuropathy), PAD, previous FU, and both previous minor and major amputations were identified as risk factors for all-cause mortality among HD subjects. However, age and retinopathy were the only risk factors for all-cause mortality among the subcohort of DM patients, and only age was predictive of CV mortality. In line with our results, a study on the characteristics of patients with DM that survived up to 11 years on HD showed that the risk of mortality increased by 3% per year increase in age [39]. Moreover, diabetic retinopathy was previously identified as an independent predictor of 3-year all-cause mortality among HD patients [40].

In our HD population, the occurrence of a new FU or amputation was more likely among patients with DM vs. non-DM, and among those with DFS vs. DM only. A history of prior FU or lower-limb amputation are conditions known to increase the risk of diabetic FU development [41,42], which was also observed in our study. However, neither the probability of all-cause nor CV mortality were predicted by previous DFS. This concords with a study conducted in Italy including a small cohort of diabetic HD subjects with critical limb ischemia and FU treated by endovascular revascularization (*n* = 99), where no variable predicted death after 12-months of follow-up [25]. Actually, HD itself is a risk factor for incident FU and amputations [16,17,43], and impaired renal function is, in turn, an independent predictor of healing failure, first amputation, and mortality [44].

The main strength of this study is that it is the first to assess mortality among DM patients with DFS on HD in our region, which included patients encompassing the entire haemodialysis population of Lleida (Spain). However, our results might not necessarily reflect those of other regions or countries with different healthcare systems or resources. For instance, mortality rates in diabetic patients with FU or amputations have been shown to vary between regions in the US, with decreased survival in those with fewer annual office visits and higher hospital admission rates [45]. Another strength is that we collected data on a wide range of comorbid conditions and chronic diabetes-related complications, such as diabetic retinopathy, reflecting the severity of microangiopathic complications of the disease. Still, the results of our study must be viewed in the context of some limitations. The main one is the retrospective nature of the design, where only association, but not causation, may be inferred from the observations. Besides, since the study was subject to pre-existing records, it is possible that not all were complete or not all potential risk factors were identified and subsequently recorded. Most importantly, we had no data on the treatment modality (i.e., conventional HD with low-flux or high-flux membranes, or hemodiafiltration), adequacy parameters (e.g., urea dialysis dose through the Kt/V value), presence of anaemia, or mineral metabolism parameters (e.g., calcium-phosphorus product or parathyroid hormone level). These variables are known to have a significant influence in HD patient’s survival and are thus potential confounding factors that were not taken into account in the study. This may have biased any association observed. Moreover, many different professionals were involved in patient care and the length of follow-up was substantial. This could have led to different measurements of the studied outcomes and risk factors throughout the database, making them less accurate and consistent than that achieved with a prospective cohort study design. Finally, we cannot discard the presence of unknown potential confounders. Another limitation was the small sample size of patients with DM and the even more reduced size of the subgroups with or without DFS. The sample size was inherently limited by the availability of subjects in the health area studied and the epidemiology of the disease. The small sample size raises the possibility that, even if there is a difference in outcomes between patients with and without DFS, we did not have enough statistical power to detect a relevant difference, and we cannot discard the absence of an actual effect (type II error). For instance, the overall mortality was very high among the DM cohort (64.1%), but similar between those with or without DFS (66.7% vs. 62.7%). Moreover, the small sample size could have also led to significant results in the absence of an actual effect (type I error). Given these limitations, the present results should be viewed as one piece of evidence, and it is necessary to perform additional studies with a larger number of subjects to detect small effects and thereby yield more significant results.

## 5. Conclusions

The major finding of our study is that DFS seems to be a condition that does not modify the pre-existing high risk of mortality among diabetic patients compared with non-diabetic subjects. These preliminary results would argue that the excess mortality attributed to DFS actually reflects the sum of chronic micro- and macrovascular diabetes-related complications, renal impairment, and HD itself. As such, there is a need for new and intensive interventions to improve the management of patients on HD with diabetes, and subsequently reduce the high mortality rates associated with the disease.

## Figures and Tables

**Figure 1 jcm-10-01368-f001:**
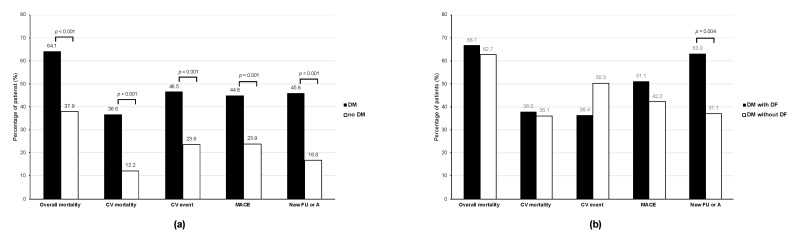
Percentage of patients on haemodialysis with studied clinical outcomes at 5-years of follow-up based on the presence of diabetes mellitus (**a**) and on the presence or absence of previous diabetic foot (**b**). CV, cardiovascular; DF, diabetic foot (ulcer or amputation); DM, diabetes mellitus; MACE, major adverse CV event.

**Figure 2 jcm-10-01368-f002:**
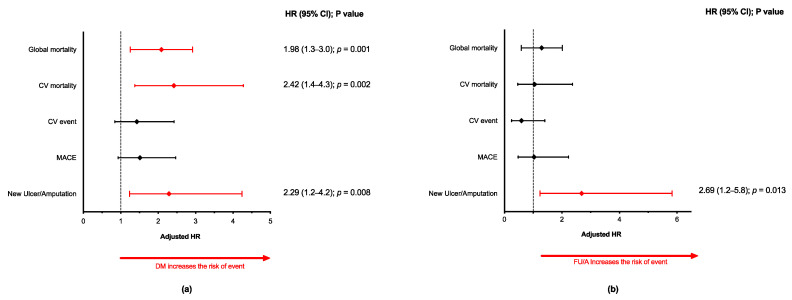
Adjusted hazard ratios for each of the studied outcomes based on the presence of diabetes mellitus (**a**) and on the presence or absence of previous diabetic foot (**b**). CV, cardiovascular; HR, hazard ratio; MACE, major adverse cardiovascular event.

**Table 1 jcm-10-01368-t001:** Baseline demographic and clinical characteristics of the study population.

			Patients with DM (*n* = 85)		*p*-Value No DM vs. DM	*p*-Value No DF vs. DF
Variable	All HD Patients	Patients without DM (*n* = 135)	No Diabetic Foot * (*n* = 55)	Diabetic Foot (*n* = 30)		
Gender, male, *n* (%)	132 (60)	80 (59.3)	31 (56.4)	21 (70.0)	0.888	0.317
Age, years, mean (SD), *n* (%)	67.5 (16.2)	66.7 (17.6)	68.5 (13.2)	69.1 (14.3)	0.332	0.856
Diabetes type, *n* (%)					-	0.508
Type 1	11 (5.0)	-	6 (10.9)	5 (16.7)		
Type 2	74 (33.6)	-	49 (89.1)	25 (83.3)		
Duration of DM, years, median (25th, 75th percentile)	19.7 (13.2; 29.7]	-	20.1 (11.5; 29.7)	18.7 (16.1; 30.7)	-	0.530
Smoking status, *n* (%)					0.037	0.206
Current	28 (17.2)	22 (16.3)	3 (5.5)	3 (10)		
Past	63 (28.6)	32 (23.7)	17 (30.9)	14 (46.7)		
Never smoked	129 (58.6)	81 (60.0)	35 (63.5)	13 (43.3)		
Hypertension, *n* (%)	189 (85.9)	117 (86.7)	47 (85.5)	25 (83.3)	0.835	0.764
Dyslipidemia, *n* (%)	107 (48.6)	46 (34.1)	41 (74.5)	20 (66.7)	<0.001	0.604
Clinical history, *n* (%)						
CHD	51 (23.2)	23 (17.0)	17 (30.9)	11 (36.7)	0.011	0.765
CVD	28 (12.7)	10 (7.4)	7 (12.7)	11 (36.7)	0.006	0.021
Cardiac arrhythmia	21 (9.5)	13 (9.6)	3 (5.5)	5 (16.7)	1.00	0.124
Heart failure	13 (5.9)	5 (3.7)	5 (9.1)	3 (10.0)	0.146	1.000
PAD, *n* (%)	110 (50.0)	51 (37.8)	37 (67.3)	22 (73.3)	<0.001	0.461
Foot ulcer, *n* (%)						
Previous	22 (10)	4 (3.0)	0 (0.0)	18 (60.0)	<0.001	<0.001
Current	20 (9.1)	4 (3.0)	0 (0.0)	16 (53.3)	<0.001	<0.001
Previous amputation, *n* (%)					<0.001	<0.001
No amputation	202 (91.8)	131 (97.0)	55 (100)	16 (53.3)		
Major	7 (3.2)	1 (0.7)	0 (0.0)	6 (20.0)		

* Diabetic foot defined as previous or current foot ulcer or amputation; CHD, coronary heart disease; CVD, cardiovascular disease; PAD, peripheral artery disease.

**Table 2 jcm-10-01368-t002:** Predictors of all-cause mortality and cardiovascular mortality in patients on haemodialysis (*n* = 220).

	All-Cause Mortality	CV Mortality
Variable	Hazard Ratio (95% CI); *p*-Value	Hazard Ratio (95% CI); *p*-Value
Gender (female)	0.69 (0.48–1.00); 0.051	0.45 (0.24–0.86); 0.015
Age (years)	1.05 (1.03–1.07); <0.001	1.04 (1.01–1.06); 0.001
DM	1.90 (1.33–2.71); <0.001	3.15 (1.76–5.63); <0.001
DM duration (years)	1.00 (0.98–1.03); 0.826	1.01 (0.98–1.04); 0.659
Hypertension	1.06 (0.63–1.79); 0.832	0.69 (0.33–1.42); 0.314
Dyslipidaemia	1.06 (0.75–1.52); 0.728	1.48 (0.83–2.62); 0.183
Smoker	0.76 (0.42–1.36); 0.353	0.79 (0.30–2.06); 0.630
CHD	1.54 (1.04–2.28); 0.031	2.19 (1.21–3.95); 0.010
CVD	1.70 (1.06–2.72); 0.027	2.52 (1.28–4.95); 0.007
Arrhythmia	1.92 (1.14–3.26); 0.015	2.28 (1.02–5.09); 0.045
HF	2.39 (1.31–4.37); 0.004	2.68 (1.05–6.84); 0.038
PAD	2.09 (1.32–3.32); 0.002	2.27 (1.10–4.68); 0.027
Diabetic retinopathy	2.32 (1.56–3.43); <0.001	4.09 (2.20–7.59); <0.001
Diabetic neuropathy	1.92 (1.34–2.76); <0.001	3.04 (1.69–5.46); <0.001
Previous FU	2.10 (1.27–3.47); 0.004	2.22 (0.99–4.95); 0.053
Current FU	1.69 (0.98–2.90); 0.058	2.12 (0.95–4.73); 0.066
Previous minor amputation	2.43 (1.27–4.67); 0.007	2.58 (0.92–7.23); 0.071
Previous major amputation	3.33 (1.54–7.21); 0.002	1.32 (0.18–9.65); 0.785

CHD, coronary heart disease; CVD, cardiovascular disease; D, diabetic; FU, foot ulcer; HF, heart failure; DM, diabetes mellitus; PAD, peripheral artery disease; T1DM, type 1 diabetes mellitus; T2DM, type 2 diabetes mellitus.

**Table 3 jcm-10-01368-t003:** Predictors of all-cause mortality and cardiovascular mortality in patients with diabetes mellitus (*n* = 85).

	All-Cause Mortality	CV Mortality
Variable	Hazard Ratio (95% CI); *p*-Value	Hazard Ratio (95% CI); *p*-Value
Gender (female)	0.72 (0.42;1.22); 0.224	0.51 (0.22;1.14); 0.102
Age (years)	1.04 (1.02;1.06); 0.001	1.04 (1.01;1.08); 0.014
DM duration (years)	1.00 (0.98;1.03); 0.826	1.01 (0.98;1.04); 0.659
Hypertension	0.90 (0.44;1.83); 0.776	0.63 (0.26;1.55); 0.313
Dyslipidaemia	0.81 (0.46;1.41); 0.451	1.63 (0.62;4.28); 0.320
Smoker	1.13 (0.40;3.20); 0.813	0.57 (0.08;4.28); 0.585
CHD	1.29 (0.76;2.19); 0.339	1.59 (0.76;3.34); 0.217
CVD	1.58 (0.89;2.80); 0.119	1.71 (0.76;3.88); 0.197
Arrhythmia	1.56 (0.71;3.43); 0.272	2.32 (0.88;6.08); 0.088
HF	1.41 (0.64;3.12); 0.399	2.06 (0.78;5.46); 0.144
PAD	1.70 (0.90;3.20); 0.101	1.44 (0.55;3.79); 0.456
Diabetic retinopathy	6.51 (1.57;27.0); 0.010	*; 0.998
Diabetic neuropathy	*; 0.997	*; 0.998
Previous FU	1.52 (0.85;2.73); 0.159	1.23 (0.50;3.01); 0.658
Current FU	1.19 (0.64;2.20); 0.575	1.37 (0.58;3.20); 0.470
Previous minor amputation	1.86 (0.84;4.12); 0.128	1.55 (0.47;5.14); 0.477
Previous major amputation	2.03 (0.86;4.78); 0.105	0.68 (0.09;5.05); 0.706

CHD, coronary heart disease; CVD, cardiovascular disease; D, diabetic; FU, foot ulcer; HF, heart failure; DM, diabetes mellitus; PAD, peripheral artery disease; T1DM, type 1 diabetes mellitus; T2DM, type 2 diabetes mellitus. * Not calculated because all the patients with the event also had diabetic neuropathy or retinopathy.

## Data Availability

The data presented in this study available on request from the corresponding author.

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
