# Peer review of "High Incidence of Adverse Outcomes in Haemodialysis Patients with Diabetes with or without Diabetic Foot Syndrome: A 5-Year Observational Study in Lleida, Spain"

_jcm, 2021, doi:10.3390/jcm10071368_

Round 1

Reviewer 1 Report

In this study, Dòria et al evaluate whether the presence of diabetic foot syndrome (DFS) is associated with an increased risk for mortality among diabetic patients undergoing routine hemodialysis (HD). To that end, the primary findings of this study are that DFS (vs no DFS) is not associated with an increased risk for mortality. Authors also examine risk of CV events, new foot ulcers, and amputations in DFS vs. no DFS patients, and also compare rates of each outcome in diabetic vs. non-diabetic HD patients. The methodology is appropriate and the paper sufficiently written.

The main strength of the study is the inclusion of all HD patients from one region in Spain with detailed clinical data. The primary limitation of this work is the relatively small sample size (n=85 patients with diabetes) which is likely underpowered for several of the outcomes examined. Indeed, the key finding is that no association was seen between DFS and mortality. Given the broader diabetes literature suggests that DFS is associated with an increased mortality risk, it is likely that study is not sufficiently powered to detect a true effect.

Further, the analysis of diabetic vs. non-diabetic HD patients with respect to all outcomes does not add any new knowledge to what we already know about the excess risk associated with diabetes in this patient population. It is unclear what or how this study contributes to our understanding of mortality risk in HD patients with or without diabetes.

Reviewer 2 Report

The study is of potential interest to the readers of the Journal, however, in my opinion, would require major revisions before publication. The retrospective design and small number of DM patients in the sample needs to be more critically discussed. The main conclusion of the study i.e. lack of impact of DFS or FU on all-cause and CV mortality, MACE or CV events is in disagreement with previously published studies. In my opinion this would require more detailed analysis in the Discussion.

The way of data presentation in the paper should be reworked. For instance Table 1 contains a lot of data, some, like "Nephropathy" unnecessary, and in current form is difficult to follow. Some of the data is repeated in the text. Figures no 3 and especially 4  are also difficult to follow and could be presented more clearly in a form of a table.

The english language needs thorough reevaluation, possibly by an english speaking person. The authors tend to use very lenghty sentences, which are difficult to understand. The use of "CVD antecedents" seems inappropriate. Page 2 line 59 "a 5-year survival rate of 1.8 years" is not clear. The definition of MACE (page 3 line 130) needs to be revised.

In conlusion I would consider the paper suitable for publication after major revision.

Reviewer 3 Report

Nice paper in which you underline an aspect still debated.  Many papers presented data attesting an higher mortality among diabetic dialized patients in the presence of DFS but on the other hand often the researchers and clinicians feeling is different.

This article has the  virtue to light the discussion.

Author Response

We thank the Reviewer for this very positive assessment.

Round 2

Reviewer 1 Report

My original concerns regarding this paper were to do with limited power to analyze primary outcomes (all-cause and CVD mortality) in people with diabetes and with vs. without a previous diagnosis of diabetic foot syndrome (DS). While the authors address this as a limitation in their revised submission, this fundamental flaw, in my opinion, does not advance our knowledge of potential effect modification by DFS status in this patient population. Further, the bulk of this paper is dedicated to examining risk factors associated with all-cause and CVD mortality in hemodialysis patients with vs. without diabetes. These findings are confirmatory, and many other studies have examined this using much larger populations. 

Author Response

We do agree that our results regarding differences between patients with or without diabetes are confirmatory and that they have to be seen only as additional evidence contributing to the existing knowledge. Moreover, we are aware that the small sample size is a major limitation in the interpretation of the results. This why, as suggested by the two Reviewers in the first peer-review round, we acknowledged that the small sample size is subject to type I and II errors, and we highlighted that the results should be interpreted with caution. In addition, we stressed that it is necessary to perform additional studies with a larger number of subjects to detect small effects and thereby yield more significant results.